# Frailty as a Moderator of the Relationship between Social Isolation and Health Outcomes in Community-Dwelling Older Adults

**DOI:** 10.3390/ijerph18041675

**Published:** 2021-02-09

**Authors:** Fereshteh Mehrabi, François Béland

**Affiliations:** 1School of Public Health (ESPUM), Université de Montréal, 7071 Parc Ave, Montréal, QC H3N 1X9, Canada; francois.beland@umontreal.ca; 2Centre de Recherche en Santé Publique (CReSP), Université de Montréal et CIUSSS du Centre-Sud-de-l’Île-de-Montréal, 7071 Parc Ave, Montréal, QC H3N 1X9, Canada; 3Lady Davis Institute for Medical Research, Jewish General Hospital, 3755, Chemin de la Côte-Ste-Catherine, Montréal, QC H3T 1E2, Canada

**Keywords:** frailty, social isolation, social networks, social support, social participation, aging

## Abstract

This research investigated the effects of social isolation on frailty and health outcomes and tested whether these associations varied across different levels of frailty. We performed a multivariate analysis of the first wave of Frailty: A longitudinal study of its expressions (FRéLE) among 1643 Canadian older adults aged 65 years and over. We assessed social isolation using social participation, social networks, and support from various social ties, namely, friends, children, extended family, and partner. Frailty was associated with disability, comorbidity, depression, and cognitive decline. Less social participation was associated with limitations in instrumental activities of daily living (IADLs), depression, and cognitive decline. The absence of friends was associated with depression and cognitive impairment. Less social support from children and partner was related to comorbidity, depression, and cognitive decline. Overall, social isolation is linked to mental health rather than physical health. The associations of having no siblings, receiving less support from friends, and participating less in social activities with ADL limitations, depression, and cognitive decline were higher among frail than prefrail and robust older adults. This study corroborates the pivotal role of social connectedness, particularly the quality of relationships, on the mental health of older adults. Public health policies on social relationships are paramount to ameliorate the health status of frail older adults.

## 1. Introduction

The effect of social isolation on health among older people has recently garnered increasing attention from the media and policymakers alike, recognizing it as an emerging public health priority [1,2]. Worldwide, roughly 50% of older people are at risk of social isolation, and about one-third of those aged 60 years and over experience loneliness in later life [3]. In Canada, one in five older adults feels socially isolated [4]. Social isolation is a known risk factor for a wide array of adverse health outcomes among older people, including disability [5], cognitive decline [6,7], depression [8], and mortality [9]. Holt-Lunstad and colleagues [10] posited that the influence of social isolation on health is comparable with that of well-established risk factors, including smoking and obesity.

In recognition of the importance of older adults’ social relationships, Berkman and Krishna [11] have developed a comprehensive conceptual model of how social networks impact health, linking social networks, social participation, and social support to health outcomes. Social networks pertain to social interactions and frequency of contact with social ties (i.e., friends, children, extended family, and partner). Emotional social support refers to the amount of love and caring provided by confident or intimate ties [11]. According to this underpinning theoretical perspective, we use a broad definition of social isolation that encompasses structural and functional aspects. The structural aspect includes social networks and social participation. The functional aspect refers to the quality of relationships or emotional social support. The impact of social isolation on health among older adults may be influenced by other factors associated with increasing age, such as frailty.

Frailty reflects the state of increased vulnerability, deriving from cumulative declines in several physiological systems [12,13]. In a landmark study, Fried and colleagues [13] proposed the “Frailty Phenotype Approach,” in which frailty leads to adverse health outcomes, including disability, comorbidity, falls, depression, cognitive impairment, and premature death [12,13]. Prior research has portrayed the link between frequent social contacts and higher social support with a lower level of frailty among older adults [2,14,15]. Researchers have suggested that frequent contact with friends [16,17,18] and neighbors [18] is more protective against frailty than contact with children. The results of a recent scoping review [2] have highlighted the link between social isolation and frailty; however, discrepancies in research results appeared when examining the effect of social isolation on adverse health outcomes. These discrepancies have led us to the assumption that frailty might moderate the association between social isolation and health outcomes, and therefore, impact this relationship differently based on the frailty status, determining which older adults are most vulnerable to poor health outcomes. Two recent studies [19,20] have investigated the combined effect of social isolation and frailty on health outcomes. The results have shown that frail and isolated older adults have a higher level of falls and mortality compared to older adults without one of these conditions or those with neither of these conditions. Nevertheless, it remains unclear whether or not frailty worsens the effect of social isolation on health. To date, a paucity of research has incorporated three dimensions of social isolation, including social participation, social networks, and social support across different types of social network ties, and little is known about the moderating role of frailty on the pathway from social isolation to health [2]. Hence, the present paper aims to investigate the effects of social isolation on frailty and adverse health outcomes and to explore how this relationship varies according to different levels of frailty. Based on the Berkman theoretical model and prior studies, this research study focuses on the following relationships:Social participation, social networks, and social support across different types of social ties are associated with frailty and adverse health outcomes.Frailty partially moderates the effects of social isolation on poor health outcomes.

From which, we derive the two following hypotheses:

**H1.** Older adults who have more contact with social ties, receive more social support, and participate more in social activities will be less frail and in better health.

**H2.** Frail and socially isolated older adults—with fewer social contacts, less social support, and lower participation in social activities—will experience higher levels of disability, cognitive decline, comorbidity, and depression than non-frail isolated older adults. This difference will be reduced among prefrail older adults and will not occur among robust older adults.

## 2. Materials and Methods

### 2.1. Data Source and Study Population

For this cross-sectional study, we employed data from the first wave of the FRéLE study (Fragilité, une étude longitudinale de ses expressions/Frailty: A longitudinal study of its expressions), a population-based study of 1643 community-dwelling men and women aged 65 years and over. Participants were recruited from a random sample of the Québec Medicare database in 2010, including a subset of three regions in the province of Québec, Canada, as follows: a metropolitan area (Montréal), a mid-sized city (Sherbrooke), and a small town (Victoriaville). The study population was stratified by gender, age groups, and study regions. Further details regarding the study sample and data collection procedures have been described in detail elsewhere [21,22]. Ethical approval for the FRéLE study was provided by the Research Ethics Committee of the Jewish General Hospital (12 January 2010). The Research Ethics Committee of the Integrated Health and Social Services University Network for West-Central Montréal (#CODIM-MBM-17-146-10 October 2020) and the Health Research Ethics Board of the Université de Montréal (#17-162-CERES-D-19-08-2020) approved the research protocol of the present study.

### 2.2. Measures

#### 2.2.1. Independent Variables

Social isolation: Based upon the Berkman theoretical model [11], we measured social isolation through participation in social activities, social networks, and receiving social support from different types of social ties, including friends, children, extended family, and an intimate partner/spouse.

Social participation was measured by 12 items, including membership in community organizations, participating in religious activities, being a volunteer, playing music, painting, visiting family members or friends, attending a community center, going to restaurants, libraries, shopping malls, cultural and sportive centers, and events [23]. Participants indicated their response on a five-point Likert scale, ranging from 1 (almost every day) to 5 (never). Scores were summed, with greater scores indicating lower social participation. The Cronbach’s alpha for this scale was 0.69.Social networks were assessed based on the longitudinal International Mobility in Aging Study’s (IMIAS) social network scale, which is a validated scale among older populations [24]. We measured social networks using the following four items: (a) the numbers of friends, living children, and extended family (i.e., grandchildren and siblings); (b) the numbers of those social ties that they see at least once a month; (c) that they have a close relationship with; and d) that they speak to by phone at least once a month [24]. The examples of questions are as follows: How many friends do you have? How many friends do you see at least once a month? How many friends do you have a very close relationship with? How many of them do you speak to by phone at least once a month? Social network questions were not asked about partners as they usually had daily contacts. Response options were “never” (code 1), “rarely” (code 2), “sometimes” (code 3), “frequently” (code 4), and “always” (code 5). The items related to each social tie were summed to give a social contact score, with higher scores indicating higher levels of social networks. The Cronbach’s alpha internal consistency estimates for friends, children, siblings, and grandchildren were 0.70, 0.87, 0.75, and 0.74, respectively.Social support was measured by the following five items of the IMIAS’s social support scale: whether participants felt helpful, loved, listened to, important to, and useful to their social ties, including friends, children, extended family, and partner [24]. The examples of questions are as follows: Do you help your friends from time to time? Do you feel that you are loved and appreciated by friends? Do your friends listen to you when you need to talk about your problems or preoccupations? Do you feel that you play an important role in your friends’ lives? Do you feel useful to your friends? The scores ranged from 1 (never) to 5 (always), with a higher score indicating a higher level of social support. The Cronbach’s alpha internal consistency estimates for friends, children, extended family, and partner were 0.72, 0.72, 0.70, and 0.73, respectively.The absence of social ties: We created a binary variable for social ties to indicate the absence of friends, children, grandchildren, siblings, and partner [25,26]. Accordingly, we dichotomized participants’ responses to the presence or absence of social ties into two categories: (a) participants with social ties (score 0) (i.e., having friends) and (b) participants without social ties (score 1) (i.e., having no friends).

#### 2.2.2. Moderator Variable

Frailty: Physical frailty was assessed based on Fried’s criteria [13], including weight loss, weakness, exhaustion, slowness, and low physical activity levels. Participants were categorized as physically frail in the presence of three or more of these criteria, as prefrail in the presence of one or two of these criteria, and as robust if none of these characteristics were observed. The detailed measurement methods for each component of frailty in the FRéLE study are provided elsewhere [27]. Frailty is described as a syndrome in the Fried phenotype of frailty. Based on the construct validity measured in the FRéLE study, frailty is a marker and determinant of health outcomes [27].

#### 2.2.3. Dependent Variables–Health Outcomes

Cognitive function was measured by the Montreal Cognitive Assessment (MoCA), which has high test–retest reliability and internal consistency. The total MoCA scores ranged from 0 to 30 points, with higher scores indicating better cognitive function (≥25) [28]. In the FRéLE’s sample population, 66 respondents had a lower cognitive status and were excluded from taking the MoCA. We censored them to the left in our analysis [21].Comorbidity was evaluated by the Functional Comorbidity Index (FCI), a validated scale that predicts older adults’ physical function [29]. Diagnoses include arthritis, osteoporosis, asthma, chronic obstructive pulmonary disease, coronary artery disease, heart failure, myocardial infarction, neurological diseases, stroke, peripheral vascular disease, diabetes, gastroduodenal pathology, depression, anxiety or panic disorders, visual impairment, hearing impairment, degenerative disc disease, obesity, and cancer. In this study, cancer was added, which was one of the comorbidities in the Cardiovascular Health Study conducted by Fried [13]. The presence of each of these conditions gave one point, with the score ranging from 1 to 19 points, with a high FCI score meaning greater comorbidity. The information on the presence of specific disease was ascertained by physician assessment.Disability was measured by the Katz [30] Index of Independence in Activities of Daily Living (ADLs) and the Lawton [31] Instrumental Activities of Daily Living (IADLs) index. ADLs include difficulty in nine self-care activities: bathing, grooming, dressing, eating, toileting, walking, getting out of bed, getting up from a chair, and cutting toenails. IADLs comprise difficulty in the nine following activities: using the telephone, using transportation, shopping, doing errands, cooking, light housekeeping, heavy housekeeping, taking medications, and managing finances. We categorized ADLs or IADLs into two groups: (1) able to perform the activity without help (score 0), and (2) unable to perform the activity (score 1). Participants who reported that they were unable to perform any of the activities were considered to have difficulty in performing ADLs or IADLs.Depressive symptoms were measured using the 15-item Geriatric Depression Scale (GDS-15) [32]. The scores ranged from 0 to 15, with greater scores suggesting greater depressive symptoms. The Cronbach alpha reliability estimate for the GDS was 0.75.

#### 2.2.4. Covariates

Covariates included demographic and socioeconomic characteristics (i.e., age, sex, education, and annual income) and life habits (i.e., smoking, alcohol consumption, and sleeping disturbance).

### 2.3. Statistical Analysis

Descriptive statistics were performed to describe the sample including means and standard deviations for continuous variables, and frequencies and percentages for categorical variables. One-way analysis of variance (ANOVA) and chi-square tests were applied to evaluate differences between frailty groups. According to the Hayes’s multi-categorical moderation model [33], we estimated frailty, a multi-categorical moderator, in the regression models by using a system of coding based on *g* − 1 variables, representing the *g* categories of frailty (*g* = 3). We thus categorized participants into frail (w1) and prefrail (w2) with reference to the non-frail group. We subsequently conducted a “slope difference test” in the moderation model to examine whether the effects of social isolation on health depended on frailty. This can be described as a test of the difference between two conditional effects of social isolation on health for two different values of frailty, including frail (w1) and prefrail (w2). As suggested by McDonough and Walters [26] and Béland and his colleagues [25], we added a binary variable for the absence of children, friends, grandchildren, siblings, and partner to all equations, considering for having or not having social ties. We performed a series of multivariate regression models to examine the effects of social isolation on frailty and health outcomes and to test whether frailty moderated the effects of social isolation on health, using Mplus version 8 [34]. We added simultaneously all dependent variables into the regression equations. In the first step, we examined our first hypothesis by testing the effects of social isolation on frailty and on adverse health outcomes, including disability, chronic diseases, depression, and cognitive decline. We then investigated our second hypothesis via examination of the interaction effects of social isolation and frailty on health outcomes. We assessed whether frailty improved model fit when added to the final model, using the Akaike Information Criterion (AIC), Bayesian Information Criterion (BIC), adjusted BIC, and chi-square tests. All multivariate regression models were controlled for covariates, and 5000 bootstrapped samples/Monte Carlo integration were performed to calculate 95% confidence intervals. The statistical significance was set at *p* ≤ 0.05.

## 3. Results

### 3.1. Participants’ Characteristics

The mean (SD) age of participants was 78.7 (7.9) years, and 50.2% of respondents were women. Almost 12.6% of participants were classified as being frail, with 38.2% being prefrail, and 49.2% robust. The level of frailty increased significantly with age. There was no gender difference between frailty groups. Frail older adults had higher levels of chronic diseases, disability, depressive symptoms, and cognitive impairment than robust ones. They had lower levels of participation in social activities, fewer social networks, and received less support from social ties. They were less educated, less likely to drink alcohol, and to have sleep disturbances. The percentage of participants who had no friends, children, grandchildren, siblings, and partner were 14.8, 14.7, 22.2, 13, and 45.5, respectively. (Table 1).

### 3.2. Social Isolation, Frailty, and Health Outcomes

Table 2 presents the results of the logistic regression of the association between social isolation and frailty. Older adults who engaged less in social activities (β: 0.595; 95% CI: 0.394, 0.789) and received less social support from children (β: −0.393; 95% CI: −0.622, −0.155) and an intimate partner (β: −0.831, 95% CI: −1.507, −0.099) were more likely to be frail. The absence of siblings (β: 0.651, 95% CI: 0.149, 1.149) was significantly associated with a higher level of frailty. However, older adults with an intimate partner (β: −1.617, 95% CI: −3.072, 0.048) and children (β: −1.297; 95% CI: −2.265, −0.245) were more likely to be frail. Our results revealed that social contacts with friends, receiving social support from friends, and having friends were not associated with frailty. Only the lack of social contact with siblings was significantly related to prefrailty (β: −0.125, 95% CI: −0.208, −0.042).

Table 3 displays the results of the association between social isolation and frailty with adverse health outcomes. It is evident from this table that frailty was associated with all poor health outcomes, including disability, depression, comorbidity, and cognitive function. Less participation in social activities was notably associated with IADLs, depression, and cognitive decline but not with ADLs and comorbidity. Less social support from children was significantly associated with comorbidity and depression. Likewise, those who received less support from extended family were at greater risk for depression. The absence of friends was associated with depression symptoms and cognitive decline. However, perceived social support from friends and social contact with friends were not linked to poor health outcomes. The presence or absence of siblings and grandchildren was unrelated to adverse health outcomes, while the presence of children was linked to depressive symptoms. Although higher levels of contact with grandchildren were related to better cognitive function; social contacts with children, siblings, and friends were not associated with older adults’ health. Further, it appears that older people who had more social contact with their grandchildren experienced a higher level of functional dependence in ADLs. Lastly, older adults who perceived less social support from a partner and had an intimate partner were more likely to be depressed or cognitively impaired.

### 3.3. The Moderating Effect of Frailty on Social Isolation and Health Outcomes

Table 4 presents the findings for the final model with interaction terms. Compared to the results of Table 3, when we added the interaction models to the previous model, the first-order coefficients for the absence of friends and the presence of a partner were no longer associated with cognitive function. The other first-order associations remained significant. The inclusion of the interaction terms improved the overall multivariate goodness of fit, according to the reduction in the AIC (from 21,811.26 to 21,794.66), and the significance of the chi-square at the 0.05 level (𝜒^2^ = 32.59). Nonetheless, the BIC and adjusted BIC values increased (from 22,259.81 to 22,286.45 and from 21,996.13 to 21,997.36, respectively), indicating that our moderation models may provide little or no extra information.

The moderation regression models in Table 4 demonstrated that the following interactions with frailty were statistically significant: social participation (β: 0.270, 95% CI: 0.071, 0.469), social support from friends (β: 0.420, 95% CI: 0.166, 0.674), having no friends (β: 1.293, 95% CI: 0.281, 2.305) and no siblings (β: 1.758, 95% CI: 0.566, 2.950). Based on the Hayes moderation model, we conducted a “slope difference test” to compare whether the effect of social isolation on health outcomes varied in different values of frailty. As presented in Table 5, the conditional effect tests showed that the negative effect of having no siblings on ADL limitations was significant for frail older adults (β: 1.242, 95% CI: 0.390, 2.094). As predicted, this effect was not apparent for prefrail and robust older adults. The subsequent conditional effects revealed that the effect of non-participation in social activities on depression was stronger for frail (β: 0.404; 95% CI: 0.119, 0.689) and prefrail (β: 0.464; 95% CI: 0.308, 0.621) older adults compared to robust ones (β: 0.194; 95% CI: 0.057, 0.331). Of importance, this effect was significantly diminished for robust older adults. Additionally, higher levels of perceived social support from friends were protective against cognitive decline for frail older adults (β = 0.323; 95% CI: 0.098, 0.547), but this benefit was significantly attenuated for prefrail and non-frail older adults. Lastly, frail older adults without friends had higher levels of cognitive decline compared to prefrail and non-frail older adults (β = 0.804; 95% CI: −0.059, 1.666). In sum, we observed that associations of having no siblings, receiving less social support from friends, and participating less in social activities with ADL limitations, cognitive decline, and depression were higher for frail older adults than for prefrail and robust ones.

## 4. Discussion

Drawing on the Berkman theoretical model of social relationships, we examined the interplay between social isolation, frailty, and health outcomes. Our results partially support our first hypothesis that older adults who engage in leisure activities, have social contacts with siblings, and perceive support from children and an intimate partner are less frail. The current study confirms the prior evidence that frailty is associated with adverse health outcomes [2]. Apart from frailty, our results indicate that actively engaging in social activities may alleviate the impact of IADL limitations, depressive symptoms, and cognitive decline among older adults. This result is consistent with evidence from previous longitudinal research [25,35] and also, is in line with the World Health Organization (WHO) framework on healthy aging [36], emphasizing the importance of social participation in later life, which may, in turn, reinforce the health of older people.

We found that older adults who perceived a shortfall in social support from children and an intimate partner were at greater risk of depression, comorbidity, and cognitive decline. The presence of an intimate partner and children and a relative lack of friends resulted in a higher likelihood of cognitive decline and depression. In this vein, our findings shed further light on the impact of intimate and kin relations on health. This interpretation is in line with previous research that emphasizes children have salient roles on the health status of Spanish and Latin American older adults [37,38]. Evidence in China and Canada yields the beneficial impact of social interactions with friends on the health of older people [37,39]. Relatedly, the findings on the importance of strong social ties for health in old age are in accord with the Berkman theory, illustrating that social ties provide essential emotional and instrumental support at times of illness [40].

Concerning social connections with different types of social ties, our results revealed that only social contacts with grandchildren were related to health outcomes. In this view, social connection with grandchildren was positively linked to better cognitive function. Contrary to expectations, our results showed that more contacts with grandchildren (a continuous variable) were associated with higher levels of independence in ADLs. As suggested by Seeman and colleagues [41], we created a binary variable, comparing those who had 0–2 grandchildren with those who had three or more grandchildren to examine whether the extreme values or gender differences were the cause of this inverse association. We ran a separate univariate analysis for males and females, entering the foregoing binary variable. The results revealed that men who had more contact with grandchildren were less likely to have ADL dependency (β = −0.453; 95% CI: 0.417, 0.969), albeit this relationship was not significant among women. This association is explained by the fact that male older adults had less functional limitations and more contact with grandchildren compared to female older adults in our sample. This binary variable was no longer significant after adjustment for covariates. The continuous variable remained significant in both univariate and multivariate analyses with a stronger association between social networks and less risk of limitations in ADL in men than in women. The results of the Survey of Health, Aging, and Retirement in Europe (SHARE) study [42] lend support to the sex difference in ADL among older adults in Northern, Eastern, and Western Europe, indicating that female older adults have a higher risk of ADL dependence than male older adults. This relationship needs further investigation in other datasets.

Taken together, our findings suggest that social isolation is linked to depression symptoms and cognitive decline rather than other adverse health outcomes in community-dwelling older adults. This result coheres with a population-based intervention in England [43], indicating that social isolation risk is related to depression and memory decline but not multiple chronic diseases and difficulties in performing ADLs and IADLs. Another longitudinal study from England [44] reached the conclusion that neither structural nor functional aspect of social relationship is associated with ADL limitations over six years. Evidence from several reviews on social isolation and health demonstrated that the most researched outcomes in physical health are mortality and cardiovascular diseases [1,45,46]. In this regard, a rapid review of 40 systematic reviews [46] found strong and consistent evidence for the association between social isolation and cardiovascular disease and depression, albeit evidence is less strong for other physical health conditions. Interventions and research studies on depression and cardiovascular diseases highlighted the absence of social support as an important risk factor for poor health outcomes, emphasizing the pivotal role of the quality of relationships [1,45].

Overall, the weak or moderate association between social isolation, frailty, and poor health outcomes is consistent with the available literature, including a scoping review of 26 studies [2], where each social relation promotes health through different mechanisms. According to this review, few studies support the impacts of both social isolation and frailty on adverse health outcomes.

Our second hypothesis pertains to the potential moderating role of frailty on the pathway from social isolation and health. Importantly, our results confirm our hypothesis that the impact of social isolation on adverse health outcomes differs depending on the frailty status. More specifically, our results revealed that the associations of receiving less support from friends and participating less in social activities with mental and cognitive impairment were stronger in frail than in prefrail and robust older adults. Hence, social isolation does not seem to promote the functional and mental health status of robust older adults but may reduce health decline in frail and prefrail older adults. Based on the recent scoping review [2], only one longitudinal study [47] has investigated the interaction effect of receiving and providing social support and frailty on mortality. The results revealed a lower risk of mortality among robust and prefrail older adults who provided social support to their family ties but not among those who received family support [2,47].

This study was cross-sectional, which limits our understanding of causative relationships between social isolation, frailty, and health outcomes. Future studies with longitudinal methods are warranted to capture developmental changes in social isolation and frailty and their effects on health outcomes over time. In particular, more research is needed to further explore the direction of the association between contact with family members and the likelihood of ADL limitations. Despite these limitations, the present study extends the social isolation domain, focusing on frailty. The notable strengths of the study include the large and population-based sample; the multicenter nature of the study; and the use of validated scales for social isolation, frailty, and health outcomes. To the best of our knowledge, this is the first attempt to focus on frailty as a moderator on the pathway from social isolation to physical and mental health, incorporating the multidimensional measure of social isolation across different types of social ties.

From a public health standpoint, the results of our study elucidate the pivotal role of kin and intimate relationships in old age, and particularly their impacts on mental and cognitive health. In this respect, several public health policies and programs implicitly incorporate social connectedness as mechanisms for enhancing older population health and well-being across the globe. As such, social participation is one of the eight domains of the Global Network of Age-Friendly Cities and Communities (AFCCs) led by the WHO in 2007. The WHO decade of Healthy Aging (2020–2030) is another initiative to promote health and well-being in later life. Several models have been developed in the United States, Canada, and Europe based on the political priorities and needs of older adults. For example, the village models of age-friendly communities [48] in the U.S. foster neighborhood social ties. In Québec, age-friendly cities [49] focus mainly on the social participation of older adults in communities, addressing social determinants of health. Despite these laudable efforts on enhancing social interrelatedness in the communities, there is scant evidence on the effectiveness of these actions and their impacts on the physical or mental health of older adults. Additionally, the current age-friendly policies focus on the physical environment but not so far on the social or mental environment [50]. At this juncture, our results underscore that social isolation influences older adults’ mental and cognitive health, though its association with physical health is notably non-statistically significant except in some limited instances. Therefore, healthcare policies and public health initiatives could benefit from considering explicitly these results in efforts aimed at reducing mental health problems and cognitive decline among vulnerable older populations. In particular, the results of our study are highly relevant for health policymakers in the context of the current coronavirus disease 2019 (COVID-19) pandemic, in which frail older adults are mostly affected by restriction measures imposed by governments all over the world. Ultimately, strategies to prevent or lessen the long-term effect of social isolation on older adults’ mental health are of paramount importance post-pandemically.

## 5. Conclusions

In conclusion, this research study is a novel contribution to the empirical literature on social gerontology by highlighting the key roles of social ties, perceived support, and engagement in social activities on promoting mental health in later life, particularly among frail older adults.

## Figures and Tables

**Table 1 ijerph-18-01675-t001:** Characteristics of the participants by frailty status.

Variables	Total (*N* = 1643)	Frail (*n* = 207)	Prefrail (*n* = 628)	Robust (*n* = 808)	*p* Value *
Age, mean (SD)	1643	84.7 (6.7)	80.4 (7.5)	75.6 (7.2)	<0.001
Age groups (%)					<0.001
65–74	536	7.7	23.2	46.3	
75–84	555	27.1	34.4	35	
85^+^	552	65.2	42.4	18.7	
Gender, (%)					0.451
Male	818	46.9	48.9	51.2	
Female	825	53.1	51.1	48.8	
Education, mean (SD)	1643	4.4 (2.7)	5.2 (2.8)	5.7 (2.8)	<0.001
Income, mean (SD)	1643	4.1 (1.7)	4.1 (1.6)	4.2 (2.7)	0.664
Smoking (%)					0.148
Current smoker	122	6.8	8.8	6.6	
Former smoker	797	44.4	46.3	51.2	
Non-smoker	724	48.8	44.9	42.2	
Alcohol (%)					<0.001
Yes	1166	48.3	67	79.8	
No	477	51.7	33	20.2	
Sleeping disturbance (%)					0.005
Yes	677	50.7	41.9	38.2	
No	966	49.3	58.1	61.8	
ADL (%)					<0.001
No difficulty	1223	32.9	69.7	88.7	
Have difficulty	420	67.1	30.3	11.3	
IADL (%)					<0.001
No difficulty	913	6.8	44.6	76.6	
Have difficulty	730	93.2	55.4	23.4	
Depression, mean (SD)	1635	5.7 (2.9)	3.4 (2.6)	1.8 (1.7)	<0.001
Comorbidity, mean (SD)	1642	4.3 (1.9)	3.6 (1.9)	2.5 (1.7)	<0.001
Cognitive function, mean (SD)	1643	19.1 (8.1)	21.9 (6.9)	24.6 (4.2)	<0.001
Social participation, mean (SD)	1643	12.6 (18.8)	17.3 (20.8)	20.7 (20.2)	<0.001
Friends					
Social network, mean (SD)	1643	12.5 (18.7)	17.3 (20.8)	20.7 (20.2)	<0.001
Social support, mean (SD)	1643	11.7 (10.5)	14.7 (9.3)	16.8 (8.2)	<0.001
No friends (%)	243	26.1	16.4	10.6	<0.001
Children					
Social network, mean (SD)	1643	10.3 (10.4)	9.4 (8.4)	8.4 (7.6)	0.005
Social support, mean (SD)	1643	14.5 (10)	16.9 (9.4)	17.3 (9.7)	<0.001
No children (%)	242	18.4	13.9	14.5	0.273
Extended family					
Social network, grandchildren, mean (SD)	1643	12.2 (14.6)	11.1 (12.8)	9.8 (11.9)	0.031
No grandchildren (%)	365	22.7	23.4	22.2	0.429
Social network, siblings, mean (SD)	1643	5.2 (7.9)	7 (7.4)	9.5 (8.4)	<0.001
No siblings (%)	214	25.1	14	9.2	<0.001
Social support family, mean (SD)	1643	15.3 (5.3)	16.9 (4.9)	17.5 (4.8)	<0.001
Partner					
Social support, mean (SD)	1643	5.3 (12.8)	9 (13.5)	11.2 (13.5)	<0.001
No partner (%)	748	59.9	47.3	40.5	<0.001

* *p* < 0.05.

**Table 2 ijerph-18-01675-t002:** Logistic regression of social isolation on frailty.

	Frailty					
Social Isolation Variables	Frail			Prefrail		
	Coefficient	CI < 0.95	CI > 0.95	Coefficient	CI < 0.95	CI > 0.95
**Intercept**	**11.111**	**7.922**	**14.156**	**3.077**	**1.210**	**4.979**
**Social participation**	**0.595**	**0.394**	**0.789**	**0.079**	**−0.022**	**0.177**
**Friends**						
Social Network	--	--	--	--	--	--
Social Support	--	--	--	--	--	--
No Friends	--	--	--	--	--	--
**Children**						
Social Network	--	--	--	--	--	--
Social Support	**−0.393**	**−0.622**	**−0.155**	**0.043**	**−0.126**	**0.218**
No children	**−1.279**	**−2.265**	**−0.245**	**0.013**	**−0.725**	**0.777**
**Extended Family**						
Social Network—Grandchildren	--	--	--	--	--	--
No Grandchildren	--	--	--	--	--	--
Social Network— Siblings	0.028	−0.140	0.180	**−0.125**	**−0.208**	**−0.042**
No siblings	**0.651**	**0.149**	**1149**	−0.285	−0.625	0.045
Social Support— Family	--	--	--	--	--	--
**Partner**						
Social Support	**−0.831**	**−1.507**	**−0.099**	−0.437	−0.936	0.051
No partner	**−1.617**	**−3.072**	**0.048**	−1.013	−2.120	0.060

Statistically significant associations are highlighted in bold. Non-statistically significant associations are indicated by two hyphens [--]. Coefficient values in plain numbers are the non-statistically significant coefficient of the categories of statistically significant independent variables. All entries are unstandardized regression coefficients. CI = confidence interval.

**Table 3 ijerph-18-01675-t003:** Regression of social isolation and frailty on health outcomes.

	ADL			IADL			Chronic Diseases			Depression			Cognitive Function		
Variables	Coef.	CI < 0.95	CI > 0.95	Coef.	CI < 0.95	CI > 0.95	Coef.	CI < 0.95	CI > 0.95	Coef.	CI < 0.95	CI > 0.95	Coef.	CI < 0.95	CI > 0.95
**Intercept**	**8.143**	**6.238**	**10.049**	**10.054**	**7.923**	**12.184**	**3.132**	**1.895**	**4.170**	**7.293**	**5.310**	**9.275**	**7.312**	**6.351**	**8.274**
**Frailty**															
Frail	**1.828**	**1.419**	**2.236**	**2.385**	**1.763**	**3.007**	**1.453**	**1.148**	**1.759**	**2.570**	**2.180**	**2.959**	**−0.567**	**−0.766**	**−0.368**
Prefrail	**0.627**	**0.321**	**0.932**	**0.653**	**0.383**	**0.923**	**0.995**	**0.796**	**1.193**	**1.045**	**0.804**	**1.285**	**−0.312**	**−0.440**	**−0.183**
**Social** **participation**	--	--	--	**0.249**	**0.122**	**0.376**	--	--	--	**0.320**	**0.217**	**0.422**	**-0.075**	**−0.131**	**−0.019**
**Friends**															
Social Network	--	--	--	--	--	--	--	--	--	--	--	--	--	--	--
Social Support	--	--	--	--	--	--	--	--	--	--	--	--	--	--	--
No Friends	--	--	--	--	--	--	--	--	--	**0.434**	**0.135**	**0.733**	**-0.274**	**−0.436**	**−0.113**
**Children**															
Social Network	--	--	--	--	--	--	--	--	--	--	--	--	--	--	--
Social Support	--	--	--	--	--	--	**−0.155**	**−0.287**	**−0.024**	**−0.363**	**−0.535**	**−0.191**	--	--	--
No Children^*^	--	--	--	--	--	--	-0.475	−1.071	0.122	**−1.423**	**−2.174**	**−0.672**	--	--	--
**Extended Family**															
Social Network Grandchildren	**0.171**	**0.050**	**0.292**	--	--	--	--	--	--	--	--	--	**0.057**	**0.001**	**0.113**
No * Grandchildren	0.050	−0.326	0.426	--	--	--	--	--	--	--	--	--	0.028	−0.131	0.186
Social Network siblings	--	--	--	--	--	--	--	--	--	--	--	--	--	--	--
No siblings	--	--	--	--	--	--	--	--	--	--	--	--	--	--	--
Social Support- Family	--	--	--	--	--	--	--	--	--	**−0.207**	**−0.328**	**−0.086**	--	--	--
**Partner**															
Social Support	--	--	--	--	--	--	--	--	--	**−0.983**	**−1.453**	**−0.513**	**0.275**	**0.028**	**0.522**
No Partner	--	--	--	--	--	--	--	--	--	**−2.007**	**−3.049**	**−0.965**	**0.592**	**0.042**	**1.141**

Statistically significant associations are highlighted in bold. Non-statistically significant associations are indicated by two hyphens [--]. * These variables should always enter the equations for considering participants without social ties. ADL: activities of daily living; IADL: instrumental activities of daily living.

**Table 4 ijerph-18-01675-t004:** Social isolation and frailty on health outcomes with interactions.

	ADL			IADL			Chronic Diseases			Depression			Cognitive Function		
Variables	Coef.	CI < 0.95	CI > 0.95	Coef.	CI < 0.95	CI > 0.95	Coef.	CI < 0.95	CI > 0.95	Coef.	CI < 0.95	CI > 0.95	Coef.	CI < 0.95	CI > 0.95
**Intercept**	**8.151**	**6.224**	**10.077**	**10.053**	**7.923**	**12.184**	**3.131**	**1.895**	**4.368**	**8.337**	**6.221**	**10.453**	**7.713**	**6.627**	**8.800**
**Frailty**															
Frail	**1.507**	**1.059**	**1.954**	**2.385**	**1.763**	**3.007**	**1.453**	**1.148**	**1.759**	**2.566**	**2.121**	**3.011**	**−0.702**	**−0.945**	**−0.459**
Prefrail	**0.602**	**0.281**	**0.924**	**0.653**	**0.383**	**0.923**	**0.995**	**0.796**	**1.193**	**1.074**	**0.833**	**1.316**	**−0.361**	**−0.533**	**−0.190**
**Social** **participation**	--	--	--	**0.249**	**0.122**	**0.376**	--	--	--	**0.194**	**0.057**	**0.331**	**−0.072**	**−0.129**	**−0.016**
**Friends**															
Social Network	--	--	--	--	--	--	--	--	--	--	--	--	--	--	--
Social Support	--	--	--	--	--	--	--	--	--	--	--	--	−0.097	−0.221	0.026
No Friends	--	--	--	--	--	--	--	--	--	**0.415**	**0.117**	**0.714**	−0.489	−1.026	0.048
**Children**															
Social Network	--	--	--	--	--	--	--	--	--	--	--	--	--	--	--
Social Support	--	--	--	--	--	--	**−0.155**	**−0.287**	**−0.024**	**−0.361**	**−0.533**	**−0.189**	--	--	--
No Children^*^	--	--	--	--	--	--	−0.474	−1.071	0.122	**−1.408**	**−2.157**	**−0.658**	--	--	--
**Extended Family**															
Social Network Grandchildren	**0.169**	**0.048**	**0.291**	--	--	--	--	--	--	--	--	--	**0.061**	**0.005**	**0.117**
No Grandchildren	0.051	−0.329	0.431	--	--	--	--	--	--	--	--	--	0.032	-0.126	0.190
Social Network sibling	--	--	--	--	--	--	--	--	--	--	--	--	--	--	--
No siblings	−0.516	−1.354	0.322	--	--	--	--	--	--	--	--	--	--	--	--
Social Support- Family	--	--	--	--	--	--	--	--	--	**−0.207**	**−0.328**	**−0.087**	--	--	--
**Partner**															
Social Support	--	--	--	--	--	--	--	--	--	**−0.998**	**−1.468**	**−0.529**	**0.252**	**0.003**	**0.501**
No Partner	--	--	--	--	--	--	--	--	--	**−2.030**	**−3.071**	**−0.990**	0.545	−0.009	1.100
**Interactions**															
**Social Participation**															
Frail	--	--	--	--	--	--	--	--	--	0.209	−0.102	0.520	--	--	--
Prefrail	--	--	--	--	--	--	--	--	--	**0.270**	**0.071**	**0.469**	--	--	--
**No Siblings**															
Frail	**1.758**	**0.566**	**2.950**	--	--	--	--	--	--	--	--	--	--	--	--
Prefrail	0.305	−0.677	1.287	--	--	--	--	--	--	--	--	--	--	--	--
**Social Support – Friends**															
Frail	--	--	--	--	--	--	--	--	--	--	--	--	**0.420**	**0.166**	**0.674**
Prefrail	--	--	--	--	--	--	--	--	--	--	--	--	0.138	−0.042	0.317
**No Friends**															
Frail	--	--	--	--	--	--	--	--	--	--	--	--	**1.293**	**0.281**	**2.305**
Prefrail	--	--	--	--	--	--	--	--	--	--	--	--	0.239	−0.527	1.006
**Summary of Model fits**															
	**LL**		**Parameters**			**AIC**			**BIC**				**Adjusted BIC**		
Model without interaction (LLh0)	−10,822.63		83			21,811.26			22,259.81				21,996.13		
Model with interaction (LLh1)	−10,806.33		91			21,794.66			22,286.45				21,997.36		
-2 (LLh0 -LLh1)	32.59		8			16.6			−26.64				−1.23		

Statistically significant associations are highlighted in bold. Non-statistically significant associations are indicated by two hyphens [--]. Coefficient values in plain numbers are the non-statistically significant coefficient of the categories of statistically significant independent variables. LL: log likelihood; AIC: Akaike’s Information Criterion; BIC: Bayesian Information Criterion.

**Table 5 ijerph-18-01675-t005:** Conditional effects of social isolation on health outcomes at different values of frailty.

	ADL				Depression			Cognitive Function		
**Social isolation indicators**	**Moderator levels**	Coef.	CI < 0.95	CI > 0.95	Coef.	CI < 0.95	CI > 0.95	Coef.	CI < 0.95	CI > 0.95
Social Participation	Frail	--	--	--	**0.404**	**0.119**	**0.689**	--	--	--
	Prefrail	--	--	--	**0.464**	**0.308**	**0.621**	--	--	--
	Robust	--	--	--	**0.194**	**0.057**	**0.331**	--	--	--
Social support-Friends	Frail	--	--	--	--	--	--	**0.323**	**0.098**	**0.547**
	Prefrail	--	--	--	--	--	--	0.040	−0.095	0.176
	Robust	--	--	--	--	--	--	−0.097	−0.221	0.026
No Friends	Frail	--	--	--	--	--	--	0.804	−0.059	1666
	Prefrail	--	--	--	--	--	--	−0.250	−0.803	0.303
	Robust	--	--	--	--	--	--	−0.489	−1.026	0.048
No Siblings	Frail	**1242**	**0.390**	**2.094**	--	--	--	--	--	--
	Prefrail	−0.211	−0.733	0.311	--	--	--	--	--	--
	Robust	−0.516	−1.354	0.322	--	--	--	--	--	--

Statistically significant associations are highlighted in bold. Non-statistically significant associations are indicated by two hyphens [--]. Coefficient values in plain numbers are the non-statistically significant coefficient of the categories of statistically significant independent variables.

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
