# Peer review of "Frailty as a Moderator of the Relationship between Social Isolation and Health Outcomes in Community-Dwelling Older Adults"

_ijerph, 2021, doi:10.3390/ijerph18041675_

Round 1
Reviewer 1 Report
My review of the manuscript “Frailty as a Moderator of the Relationship Between Social Isolation and Health Outcomes in Community-Dwelling Older Adults ” is very positive. It covers an interesting topic through a rigorous scientific study. My decision is that the article should be accepted but there are minor corrections that should be considered in the review process.
Namely:
Methods:
- Please indicate the meaning of “FRéLE”.
- It was not clear for me if the social networks and support and the absence of social ties was evaluated through an validated instrument or with single itens. Please clarify this aspect, for instance referring the instrument, the number and example of items, and the way the variable was coded.
- Also in this section, about the absence of social ties, instead of presenting the Cronbach’s alpha, data about the percentage of participants who had no friends, children, grandchildren, siblings, and partner was presented. This data is a result of the study, not information on validity, so it doesn't make sense to be in the methodology.
- Please be clearer if the 1,643 community-dwelling men and women aged 65 years and over included the 66 who were excluded because of lower cognitive status.
Results:
- I am surprised by the low representation of women in your sample since in most studies with older adults samples the women are in a greater number. Any explanation?
- In table 1 it makes more sense to present the sociodemograhic data in the first lines (previous to lifestyle variables).
- Please indicate how the level of frailty was significantly different among the age groups.
- Table 2 – if the Significant associations are highlighted in Bold and the Non-significant associations are indicated by two hyphens [--], what are the numbers without bold? Please clarify.
- Also here, in the Logistic regression of social isolation on frailty, why did the authors presente both CI<0.95 and CI>0.95 values?
- Tables 3, 4 and 5 – please indicate the meaning of bold.
Overall:
- More attention to detail errors; for instance on table 2 the coef. of social support from partner is -0.831 but in the text is -0.832.
- On discussion author refer “Based on our published scoping review [2]”; I think the “our” is not necessary because of self-citations.
Author Response
Response to Reviewer 1 Comments
My review of the manuscript “Frailty as a Moderator of the Relationship Between Social Isolation and Health Outcomes in Community-Dwelling Older Adults ” is very positive. It covers an interesting topic through a rigorous scientific study. My decision is that the article should be accepted but there are minor corrections that should be considered in the review process.
We are grateful to the reviewer for providing constructive and helpful feedback on our paper. We have been able to incorporate changes to reflect most of the suggestions provided by the reviewer. Our point-by-point responses are provided below. We uploaded a copy of the article with the changes tracked.
Methods:
Point 1: Please indicate the meaning of “FRéLE”.
Response 1: We defined this abbreviation, which is originally in French and its English equivalent (Fragilité, une étude longitudinale de ses expressions/ Frailty: A longitudinal study of its expressions) (Line 97).
Point 2: It was not clear for me if the social networks and support and the absence of social ties was evaluated through a validated instrument or with single items. Please clarify this aspect, for instance referring the instrument, the number and example of items, and the way the variable was coded.
Response 2: We used the longitudinal International Mobility in Aging Study’s (IMIAS) Social Network and Social Support scales which has high test-retest reliability along with internal consistency among older populations (Cronbach alpha coefficient ≥ 0.80) (Ahmed et al., 2018, reference number: 24). To avoid ambiguity, we referred to the IMIAS scales and also, added the number of items for social networks (4 items), and social support (5 items) scales. Accordingly, we gave examples of questions and the way that variables were coded [lines 125 and 160, pages 3-4].
Point 3: Also in this section, about the absence of social ties, instead of presenting the Cronbach’s alpha, data about the percentage of participants who had no friends, children, grandchildren, siblings, and partner was presented. This data is a result of the study, not information on validity, so it doesn't make sense to be in the methodology.
Response 3:
We agree and have moved the percentages to the result sections (lines 232-234). For four items of the social network scale and 5-items of the social support scale, we had participants who had no friends, children, siblings, grandchildren, and partner, and consequently, they did not respond to any questions about these social ties. we created a binary variable for having and not having social ties to control these zero values. Thus, there is no Cronbach alpha for these binary variables.
Point 4: Please be clearer if the 1,643 community-dwelling men and women aged 65 years and over included the 66 who were excluded because of lower cognitive status.
Response 4: Yes, the sample included 66 participants. In the description of the FRéLE sample population is mentioned that “no cognitive exclusion criterion was applied for recruiting the participants” (Provencher et al, 2017, reference number: 22). We clarified the issue in the cognitive function measurements as follows: “in the FRéLE’s sample population, 66 respondents had a lower cognitive status” (Line 177).
Results:
Point 5: I am surprised by the low representation of women in your sample since in most studies with older adults samples the women are in a greater number. Any explanation?
Response 5: The FRéLE sample is stratified into 6 groups as follows: men and women and three age groups (65-74, 75-84, and 85+). Thus, an equal number of individuals is selected in each group. The detail on the stratification strategy is provided in Provencher and colleagues article (2017) [reference number: 22].
Point 6: In table 1 it makes more sense to present the sociodemographic data in the first lines (previous to lifestyle variables).
Response 6: As suggested, we presented the sociodemographic variables at the first lines in Table 1.
Point 7: Please indicate how the level of frailty was significantly different among the age groups.
Response 7: We added the age groups in Table 1 and the following description in the results section: “The level of frailty increased significantly with age” (line 241).
Point 8: Table 2 – if the Significant associations are highlighted in Bold and the Non-significant associations are indicated by two hyphens [--], what are the numbers without bold? Please clarify.
Response 8: We added the explanation in the Tables’ notes and statistical analysis.
Point 9: Also here, in the Logistic regression of social isolation on frailty, why did the authors present both CI<0.95 and CI>0.95 values?
Response 9: It is necessary to report the lower and upper values of the confidence intervals (CI) to ensure that a null hypothesis test to the level of 5% cannot be rejected.
Point 10: Tables 3, 4 and 5 – please indicate the meaning of bold.
Response 10: As suggested, we added the meaning of Bold and hyphens in the Tables’ notes.
Overall:
Point 11: More attention to detail errors; for instance, on table 2 the coef. of social support from partner is -0.831 but in the text is -0.832.
Response 11: Thanks for pointing this out. We corrected the errors.
Point 12: On discussion author refer “Based on our published scoping review [2]”; I think the “our” is not necessary because of self-citations.
Response 12: We modified it into “Based on the recent scoping review”.

Reviewer 2 Report
This a well written and interesting study on the association between frailty and different health outcomes.
As described by the authors, the cross-sectional design limits the possibilities to describe causal associations and true predicitve relations between frailty and the outcomes, which has been shown in other longitdonal studies.
Even though the authors in general correctly use the term "association", there are examples (line 235) where the more stronger term "predictor" was used. Since the design of the study does not show true prediction of outcome, I suggest the authors carefully go through the ms and consistantly use the term "association".
Author Response
Response to Reviewer 2 Comments
This a well written and interesting study on the association between frailty and different health outcomes.
We are grateful to the reviewer for providing constructive and helpful feedback on our paper. Our response is provided below. We uploaded a copy of the article with the changes tracked.
Point 1: As described by the authors, the cross-sectional design limits the possibilities to describe causal associations and true predictive relations between frailty and the outcomes, which has been shown in other longitudinal studies. Even though the authors in general correctly use the term "association", there are examples (line 235) where the more stronger term "predictor" was used. Since the design of the study does not show true prediction of outcome, I suggest the authors carefully go through the ms and consistently use the term "association".
Response 1: Thank you for pointing this out. We agree and have modified the terms ” predictor variables /predicted” to ” variables/associated” throughout the manuscript and tables.
